# Anatomical Characteristics of Cervicomedullary Compression on MRI Scans in Children with Achondroplasia

**DOI:** 10.3390/jimaging10110291

**Published:** 2024-11-14

**Authors:** Isabella Trautwein, Daniel Behme, Philip Kunkel, Jasper Gerdes, Klaus Mohnike

**Affiliations:** 1Children’s Hospital, Otto-von-Guericke-University, 39120 Magdeburg, Germany; isa.trautwein@gmx.de (I.T.); jasger99@gmail.com (J.G.); 2Clinic for Neuroradiology, University Hospital, 39120 Magdeburg, Germany; daniel.behme@med.ovgu.de; 3Department of Pediatric Surgery, University Medical Centre Mannheim, 68167 Mannheim, Germany; philip.kunkel@umm.de

**Keywords:** achondroplasia, magnetic resonance imaging, cervicomedullary compression, foramen magnum stenosis, decompression surgery

## Abstract

This retrospective study assessed anatomical characteristics of cervicomedullary compression in children with achondroplasia. Twelve anatomical parameters were analyzed (foramen magnum diameter and area; myelon area; clivus length; tentorium and occipital angles; brainstem volume outside the posterior fossa; and posterior fossa, cerebellum, supratentorial ventricular system, intracranial cerebrospinal fluid, and fourth ventricle volumes) from sagittal and transversal T1- and T2-weighted magnetic resonance imaging (MRI) scans from 37 children with achondroplasia aged ≤ 4 years (median [range] 0.8 [0.1–3.6] years) and compared with scans from 37 children without achondroplasia (median age 1.5 [0–3.9] years). Mann–Whitney U testing was used for between-group comparisons. Foramen magnum diameter and area were significantly smaller in children with achondroplasia compared with the reference group (mean 10.0 vs. 16.1 mm [*p* < 0.001] and 109.0 vs. 160.8 mm^2^ [*p* = 0.005], respectively). The tentorial angle was also steeper in children with achondroplasia (mean 47.6 vs. 38.1 degrees; *p* < 0.001), while the clivus was significantly shorter (mean 23.5 vs. 30.3 mm; *p* < 0.001). Significant differences were also observed in myelon area, occipital angle, fourth ventricle, intracranial cerebrospinal fluid and supratentorial ventricular volumes, and the volume of brainstem protruding beyond the posterior fossa (all *p* < 0.05). MRI analysis of brain structures may provide a standardized value to indicate decompression surgery in children with achondroplasia.

## 1. Introduction

Achondroplasia is the most common form of non-lethal skeletal dysplasia, occurring in approximately 1 in 25,000–30,000 live births worldwide [1,2,3,4]. It is characterized by a disproportionate, short stature, a disproportionately large head (macrocephaly), midface and skull base hypoplasia, shortening of the proximal limbs (rhizomelia), and other skeletal abnormalities which can lead to medical, functional, and psychosocial problems throughout life [5]. Achondroplasia is an inherited genetic disorder caused by autosomal dominant gain-of-function pathogenic variants in the fibroblast growth factor receptor 3 gene (FGFR3) located on the short arm of chromosome 4 [6,7]. Overactivation of FGFR3 affects endochondral ossification and leads to reduced bone growth [8].

Foramen magnum stenosis (FMS) is one of the most common and serious complications of achondroplasia in infancy and early childhood [9]. The foramen magnum resides in the occipital bone, within the posterior cranial fossa, and is traversed by various structures including the medulla oblongata, radix spinalis of the nervus accessorius, and arteria vertebrales and spinales [10,11,12]. In babies with achondroplasia, the foramen magnum is smaller than average; restricted growth in the first 2 years of life and premature closure of the skull base synchondroses further impact foramen magnum size and shape [13,14]. A small and misshapen foramen magnum can result in compression of the brainstem and spinal cord, leading to frequent apnea episodes which greatly increase the risk of sudden infant death syndrome in infancy [15,16]. Possible signs of FMS include severe hypotonia, motor delays, feeding and sleep disturbances, and clinical features of myelopathy such as hyperreflexia [17,18]. Rates of mortality due to FMS in achondroplasia are 2–7.5% [19,20], which represents much higher rates of mortality than the general population [20,21,22]. In the absence of established guidelines, craniocervical decompression surgery rates in patients with achondroplasia vary significantly across centers, ranging from 6.3% to 40% [13,23,24]. In 2021, a simple scoring system for categorizing the severity of FMS on magnetic resonance imaging (MRI) was developed [25]. The objective of the Achondroplasia Foramen Magnum Score (AFMS) is to detect spinal cord compression in children with achondroplasia as early as possible through routine MRI screening. This helps to ensure timely intervention, thereby reducing morbidity and mortality as a consequence of FMS. In children with severe FMS, decompression surgery is the accepted treatment modality, with the aim of alleviating the impingement of the dural sac and spinal cord. In asymptomatic patients, there is no consistent approach to when surgery should or should not be performed [5].

In 1985, Cohen et al. [26] analyzed craniofacial morphology from lateral X-ray cephalograms of 51 achondroplastic individuals using 66 linear, angular, and area variables. In comparison with 951 average-stature adults, significant findings included shortened posterior cranial base length and acute cranial base angle. The authors interpreted the observed findings by earlier closure of the intersphenoidal synchondrosis [26]. Over the last few decades, MRI-based quantitative morphovolumetry has allowed a detailed approach to measure areas, volumes, and lengths of other parameters, including the clivus and supraocciput or tentorial angles [27].

These anatomical landmarks provide an informative dataset to be used for the development of deep learning (DL) systems. We therefore investigated MRI scans in children with achondroplasia compared with a healthy reference group. The children with achondroplasia were further subdivided into those who received decompression surgery for FMS and those who did not. The overall aim of the study was to build upon the AFMS to identify further anatomical characteristics in children with achondroplasia that may allow development of (1) a standard indication for decompression surgery and (2) a standardized evaluation of effect of medical treatment on critical intracranial structures.

## 2. Materials and Methods

### 2.1. Patient Population

Data collection for this retrospective study took place between October 2020 and April 2023 at Magdeburg University Hospital, Magdeburg, Germany. Children aged ≤ 4 years with confirmed achondroplasia who had received a skull base MRI scan were identified from the Germany-wide CrescNet Registry [28]. A reference population of children aged ≤ 4 years without achondroplasia who had received a skull base MRI scan were identified from Magdeburg University Hospital records. The reference population could have no evidence of cervicomedullary damage and no diseases that could have influenced associated anatomical structures.

MRI files were made available by the treating physicians, parents or guardians of the children for a second evaluation. The study was conducted in accordance with the Declaration of Helsinki and the opinion of the Ethics Committee of the Magdeburg University Hospital Medical Faculty, whereby an evaluation of stored data is permitted in accordance with the doctor–patient contract, the General Contractual Conditions (AVB) for the University Hospital Magdeburg, Institution under Public Law (A.ö.R) i. d. a. F., via § 16 (5). Patient consent was waived by the ethics committee due to the anonymized nature of the data.

### 2.2. Analysis of Anatomical Parameters

MRI images from the achondroplasia and reference groups were analyzed using Dornheim Segmenter Research software (v2019.1; Dornheim Medical Images GmbH, Magdeburg, Germany) [29]. To segment the region of interest (ROI), the investigator first annotated the image by drawing a bounding box. Thereafter, the software automatically calculated areas, volumes, lengths, and angles within the images. Only preoperative MRI scans were included in the study; if multiple scans were available, only the first MRI scan of each child was used.

The following 12 parameters were measured using T1- and T2-weighted images in sagittal and transversal planes: diameter of the foramen magnum (T1 sagittal); area of the foramen magnum; area of the myelon (T1 transversal); length of the clivus (from the dorsum sellae to the end of the clivus); tentorium angle (the angle between the highest point of the tentorium, the protuberantia occipitalis interna and the dorsum sellae in the mid-sagittal plane); occipital angle; volume of the posterior fossa; proportion of brainstem volume outside the posterior fossa; volume of the cerebellum (T2 sagittal); volume of the supratentorial ventricular system (including the third ventricle as well as the lateral ventricles); volume of the intracranial cerebrospinal fluid (CSF) system (T2 transversal); and volume of the fourth ventricle (T2 sagittal). These parameters were selected from previous observational studies [26,27] and based on personal experience of a board-certified neurosurgeon (PK) with sub-specialization in surgical treatment of conditions of the cranial cervical junction (CCJ).

Before each measurement, the selected structure and the quality of the MRI image were reviewed; images of insufficient quality and structures that were incomplete were excluded. The brightness and contrast of the image were optimized to better distinguish the structures to be measured and enable more accurate measurements to be taken by the software. Each measurement was conducted in duplicate per MRI scan to compensate for any measurement errors.

### 2.3. Calculation of AFMS

The AFMS was calculated according to the method described previously [25]. Briefly, sagittal and axial T2-weighted MRI images from the achondroplasia group were scored from 0 (no evidence of FMS) to 4 (severe FMS) based on the degree of foramen magnum narrowing and spinal cord changes.

### 2.4. Statistical Analysis

Anatomical parameters measured from MRI scans are presented as mean (standard deviation) and median (range). Parameters were compared between the achondroplasia and reference groups using a Mann–Whitney U test for independent samples based on a 0.05 significance level.

## 3. Results

MRI images from 37 patients (20 males and 17 females) with confirmed achondroplasia were analyzed and compared with MRI scans from 37 individuals (23 males and 14 females) without achondroplasia. Within the achondroplasia group, 13 patients (6 females and 7 males) had received decompression surgery of the foramen magnum.

The mean age of the children with achondroplasia at the time of MRI was 1.1 years (1.1 years in those who had received decompression surgery and 0.9 years in those who had not received decompression surgery), while the mean age of the reference group was 1.7 years (Table 1). The oldest patient at the time of MRI across both groups was 3.9 years.

Illustrative examples of key anatomical structures on MRI scans from children with achondroplasia are shown in Figure 1 and Figure 2. Of the 12 parameters measured and analyzed from the MRI scans, 10 showed significant differences between the achondroplasia and reference groups (Figure 3; Appendix A). The diameter (mean 10.0 mm vs. 16.1 mm; *p* < 0.001) and area (mean 109.0 mm^2^ vs. 160.8 mm^2^; *p* = 0.005) of the foramen magnum were significantly smaller in children with achondroplasia compared with the reference group. Furthermore, the myelon area was significantly smaller (mean 40.6 mm^2^ vs. 47.7 mm^2^; *p* = 0.004) and the clivus significantly shorter (mean 23.5 mm vs. 30.3 mm; *p* < 0.001) in children with achondroplasia. The tentorium angle was significantly steeper in children with achondroplasia (mean 47.6 degrees vs. 38.1 degrees; *p* < 0.001) and was accompanied by a larger volume of “overhang” of the brainstem from the posterior cranial fossa (mean 4542.5 mm^3^ vs. 2614.4 mm^3^; *p* < 0.001). A significantly smaller volume of the fourth ventricle (mean 749.9 mm^3^ vs. 1056.2 mm^3^; *p* = 0.029) and corresponding significantly larger volume of the supratentorial ventricular system (mean 42,062.3 mm^3^ vs. 13,389.2 mm^3^; *p* < 0.001) was also observed in patients with achondroplasia. No significant differences between the achondroplasia and reference groups were observed in posterior fossa or cerebellum volume.

When comparing children with achondroplasia who had received decompression surgery with those who had not, the diameter of the foramen magnum showed a significant difference between groups (Table 2; *p* < 0.05).

AFMS could be retrospectively determined in 24 of the 37 children with achondroplasia (Figure 4). Of these, 9 children underwent decompression surgery, of whom 6 (66.7%) had an AFMS score of 3 or 4. Of the 15 children who did not undergo decompression surgery, 14 had an AFMS score of 1 or 2 (93.3%). Measurements of the diameter of the foramen magnum, the clivus length, the tentorial angle, the fourth ventricle volume, the supratentorial CSF system volume, and the area of “overhang” of the brainstem from the posterior cranial fossa were consistent with the AFMS (Table 3).

## 4. Discussion

Cervicomedullary compression and FMS are well recognized complications of achondroplasia, with infants and younger children at higher risk of these complications than adults and older children [5,25]. Certain measures, such as the AFMS, have been developed to aid identification of early spinal cord changes and grade severity of FMS. However, identifying candidates for decompression surgery remains an area of inconsistency [30,31,32,33]. In addition, a new medical treatment for achondroplasia, the recombinant C-type natriuretic peptide analogue vosoritide, has recently shown improved growth of facial volume, sinus volume, and foramen magnum area in achondroplastic infants [34]. Therefore, further proof of MRI changes induced by vosoritide or investigational products may help guide the decision to prescribe treatment.

The present study aimed to identify further anatomical characteristics in children with achondroplasia that may allow a standard indication for decompression surgery. Of 12 parameters defined and measured in MRI images from children with achondroplasia, 10 showed significant differences compared with the reference population. Some of the differences identified were consistent with previous studies in children with achondroplasia. The reduction in foramen magnum area and diameter observed in the present study has been well characterized previously in children with achondroplasia [14,25]. However, the postulated link between foramen magnum changes and a shorter posterior cranial fossa due to underdevelopment of the occipital and clivus bone components as a consequence of defective endochondral bone formation and premature fusion of posterior fossa synchondrose [35,36,37,38] was not confirmed in our study. This may have been related to a disparity in the number of children with complete imaging of the posterior fossa (n = 24 in the achondroplasia group vs. n = 37 in the reference group). Our study was also consistent with previous studies showing a shortening of the clivus in achondroplasia [35]. The underlying reason for clivus shortening might be the early fusion of the spheno-occipital synchondrosis [27]. A correlation between the length of the clivus and the supratentorial ventricular system, caused by either a CSF flow modeling effect during development or an early closure of spheno-occipital synchondrosis leading to the characteristic concave shape of the shortened clivus has been shown previously [27,36,39,40]. However, in contrast to previous results [27], our results also revealed changes in the fourth ventricle and the supratentorial ventricular system. While the fourth ventricle was smaller in children with achondroplasia, the volume of the supratentorial ventricular system was increased. Longer-term studies assessing the dynamics of cerebrospinal fluid could help elucidate the mechanisms underlying this observation. The alignment of the tentorial angle in children with achondroplasia has been characterized extensively in previous studies [27,35]. Accordingly, a significantly steeper angle in children with achondroplasia was observed in our study versus the reference group. It is noteworthy that the tentorial angle represented one of the borderlines in the measurement of the posterior fossa, the volume of which, in turn, determines the “overhang” of the brainstem beyond the posterior fossa. As a result, the increased volume of the overhang coincided with the steeper tentorial angle in children with achondroplasia versus the reference group. An increased tentorial angle caused by vertical orientation of the cerebellar tentorium supports the view that the tentorium is pushed forward and becomes steeper in children with achondroplasia to compensate for the undersized cranial base bone and abnormal growth of the neural structures of the posterior fossa [27]. A relative increase in volume of the brainstem may occur in achondroplasia, a disorder in which the skull base bone is primarily affected [41]. However, this could not be precisely demonstrated in our study as only the bony rim of the posterior fossa and the overload were measured, not the neural structures in the posterior fossa.

The AFMS was developed to categorize the severity of FMS on MRI, including measurements of the degree of narrowing of the foramen magnum, preservation of the CSF and evidence of spinal cord compression [25]. Eligibility for decompression surgery is based on evidence of spinal cord compression in the absence of spinal cord signal changes (AFMS3), or evidence of both spinal cord compression and spinal cord signal changes (AFMS4) [25]. Retrospectively applying AFMS criteria to our achondroplasia cohort indicated that the decision to conduct decompression surgery was made correctly in 66.3% of those operated upon, based on those who would have scored 3 or 4 on preoperative MRI scans; 93.3% of patients with an AFMS score of 1 or 2 did not receive surgery. In the present study, six of the 12 parameters measured on MRI scans aligned with AFMS when the latter scores were considered in their totality. This was not surprising, as these parameters overlapped with those that showed significant differences between children with achondroplasia and the reference population.

This study was initiated to assess anatomical differences in intracranial structures in patients with achondroplasia, to interpret advanced imaging modalities. Ten out of twelve parameters showed significant differences in patients with achondroplasia compared with the reference population. Therefore, MRI offers a great opportunity to develop a DL tool to assist in (1) the indication of decompression surgery and (2) to measure the outcome following vosoritide or other precision therapies.

The present study had some limitations. The sample size was small, with 37 patients in both the achondroplasia and reference groups, and only 14 and 23 patients, respectively, in the operated and non-operated subgroups. While the rarity of achondroplasia precluded enrollment of a larger cohort of infants of the same age with a preoperative MRI, a strength of our study was the inclusion of patients from across Germany via the CrescNet Registry. Future studies could perhaps be conducted in an international setting to include a larger number of patients and increase the validity of the data. Due to the retrospective study design, some MRI sequences were missing or the anatomical structure under investigation was not pictured fully; therefore, not every parameter could be measured in all children. In addition, the use of the Dornheim Segmenter software [29] was associated with some limitations during measurement. For example, poor focus in some MRI images may have prevented reliable identification of certain structures. Steps were taken to maximize inclusion of MRI images from our cohort, including optimization of image brightness and contrast to better distinguish the structures to be measured and enable more accurate measurement by the software. Each measurement was also conducted in duplicate to compensate for any measurement errors.

## 5. Conclusions

In conclusion, MRI analysis of anatomical structures such as the clivus, supratentorial ventricular system, tentorium angle, and brainstem protrusion, in combination with the AFMS, may provide a standard indication for decompression surgery. Further investigation is needed to improve understanding of the correlation between the anatomical structures and clinical presentation of children with achondroplasia. These specific MRI findings offer a great opportunity to develop a DL tool that will assist in (1) the indication of decompression surgery and (2) measuring the outcome following vosoritide or other precision therapies.

## Figures and Tables

**Figure 1 jimaging-10-00291-f001:**
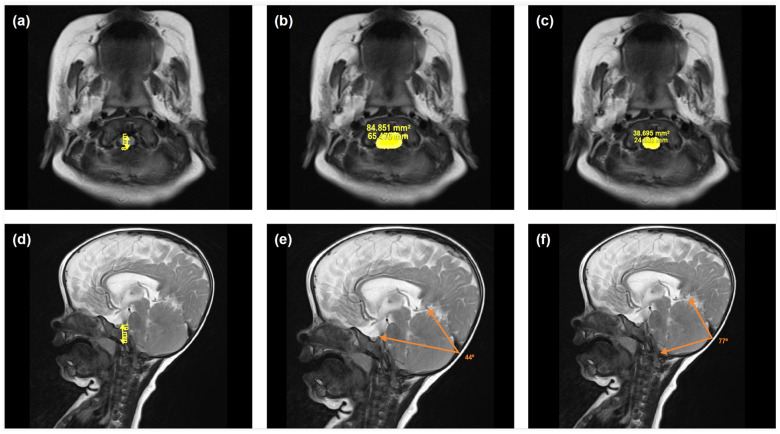
Illustrative examples of selected anatomical structures measured from magnetic resonance imaging scans from a child with achondroplasia. Shown are images used to calculate the following parameters: (**a**) foramen magnum diameter, (**b**) foramen magnum area, (**c**) myelon area, (**d**) clivus length, (**e**) tentorium angle, and (**f**) occipital angle.

**Figure 2 jimaging-10-00291-f002:**
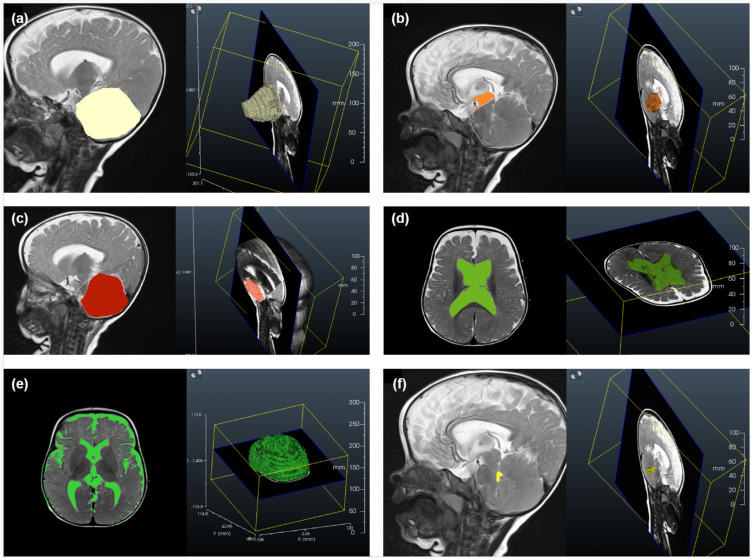
Illustrative examples of selected anatomical structures measured from magnetic resonance imaging scans from a child with achondroplasia, including screenshots of the data segmentation process using Dornheim Segmenter Research software (v2019.1; Dornheim Medical Images GmbH, Magdeburg, Germany). 3-D reconstructions of the following segmented structures are shown: (**a**) posterior fossa volume, (**b**) proportion of brainstem volume outside the posterior fossa, (**c**) cerebellum volume, (**d**) supratentorial ventricular system volume, (**e**) intracranial cerebrospinal fluid system volume, and (**f**) fourth ventricle volume.

**Figure 3 jimaging-10-00291-f003:**
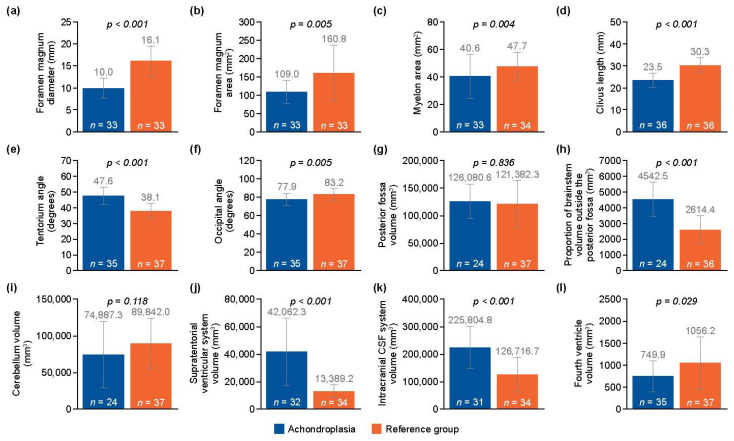
Analysis of the following anatomical parameters on magnetic resonance imaging scans in children with achondroplasia compared with the reference group: (**a**) foramen magnum diameter, (**b**) foramen magnum area, (**c**) myelon area, (**d**) clivus length, (**e**) tentorium angle, (**f**) occipital angle, (**g**) posterior fossa volume, (**h**) proportion of brainstem volume outside the posterior fossa, (**i**) cerebellum volume, (**j**) supratentorial ventricular system volume, (**k**) intracranial cerebrospinal fluid (CSF) system volume, and (**l**) fourth ventricle volume. Mean (standard deviation [SD]) data are shown. *p*-values are from Mann–Whitney U tests for independent samples.

**Figure 4 jimaging-10-00291-f004:**
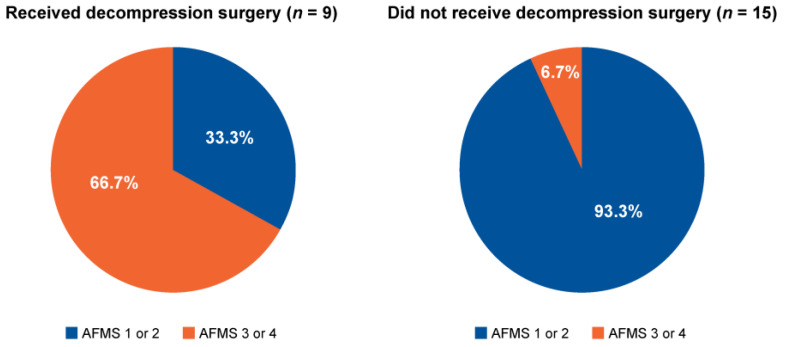
Proportion of patients who did/did not receive decompression surgery who were retrospectively graded as candidates for decompression surgery based on application of the Achondroplasia Foramen Magnum Score (AFMS) to magnetic resonance imaging scans.

**Table 1 jimaging-10-00291-t001:** Baseline characteristics.

Characteristic	Achondroplasia (*N* = 37)	Reference (*N* = 37)
Sex, *n* (%)		
Male	20 (54.1)	23 (62.2)
Female	17 (45.9)	14 (37.8)
Age, years ^a^		
Mean (SD)	1.1 (0.9)	1.7 (1.3)
Median (range)	0.8 (0.1–3.6)	1.5 (0–3.9)
Decompression surgery, *n* (%)		
Yes	13 (35.1)	0
No	23 (62.2)	37 (100)

^a^ At time of magnetic resonance imaging.

**Table 2 jimaging-10-00291-t002:** Analysis of anatomical parameters on magnetic resonance imaging scans in children with achondroplasia who received decompression surgery compared with those who did not receive surgery.

Parameter		Received Surgery	Did Not Receive Surgery	*p*-Value *
Foramen magnum diameter, mm	No. patients	12	21	<0.1
	Mean (SD)	8.6 (1.9)	10.7 (2.2)	
	Median (range)	8.5 (5.5–12.5)	10.5 (7.5–16.6)	
Foramen magnum area, mm^2^	No. patients	12	21	0.922
	Mean (SD)	108.3 (31.5)	109.4 (32.2)	
	Median (range)	118.8 (49.6–155.4)	111.8 (54.9–163.8)	
Myelon area, mm^2^	No. patients	12	21	0.228
	Mean (SD)	36.0 (14.2)	43.3 (16.8)	
	Median (range)	33.9 (16.5–66.9)	39.3 (25.2–102.3)	
Clivus length, mm	No. patients	14	22	0.602
	Mean (SD)	23.2 (3.2)	23.7 (3.1)	
	Median (range)	22.6 (19.5–32.5)	23.3 (16.0–30.0)	
Tentorium angle, degrees	No. patients	14	21	<0.001
	Mean (SD)	49.0 (4.4)	46.6 (6.0)	
	Median (range)	49.0 (43.5–57.0)	47.0 (31.5–58.0)	
Occipital angle, degrees	No. patients	14	21	0.513
	Mean (SD)	79.1 (5.8)	77.0 (7.1)	
	Median (range)	79.5 (70.0–89.5)	78.5 (64.0–88.0)	
Posterior fossa volume, mm^3^	No. patients	7	17	0.120
	Mean (SD)	110,595.6 (20,307.3)	132,456.8 (32,772.2)	
	Median (range)	106,081.3 (84,624.6–140,277.7)	134,941.4 (48,439.7–177,870.0)	
Proportion of brainstem volume outside the posterior fossa, mm^3^	No. patients	7	17	<0.001
	Mean (SD)	4698.3 (1034.4)	4478.3 (1136.4)	
	Median (range)	4532.0 (3233.8–6301.0)	4428.4 (2778.6–7021.3)	
Cerebellum volume, mm^3^	No. patients	7	17	0.782
	Mean (SD)	90,139.4 (16,998.3)	68,607.0 (51,701.3)	
	Median (range)	86,698.1 (68,759.5–113,405.8)	84,353.8 (98.7–141,281.0)	
Supratentorial ventricular system volume, mm^3^	No. patients	12	20	<0.001
	Mean (SD)	43,054.2 (28,955.2)	41,467.1 (21,845.7)	
	Median (range)	33,671.7 (16,830.3–115,394.5)	35,651.9 (10,782.3–108,493.8)	
Intracranial CSF system volume, mm^3^	No. patients	12	19	<0.001
	Mean (SD)	221,020.3 (86,479.5)	228,826.5 (71,079.6)	
	Median (range)	189,483.0 (124,900.2–376,599.0)	236,400.2 (118,999.2–337,691.4)	
Fourth ventricle volume, mm^3^	No. patients	14	21	0.797
	Mean (SD)	716.1 (318.5)	772.5 (383.8)	
	Median (range)	732.0 (292.0–1279.2)	660.3 (319.6–1607.8)	

* Mann–Whitney U test for independent samples.

**Table 3 jimaging-10-00291-t003:** Analysis of anatomical parameters on magnetic resonance imaging scans in children with achondroplasia by Achondroplasia Foramen Magnum Score (AFMS).

Parameter		AFMS 1 or 2	AFMS 3 or 4
Foramen magnum diameter, mm	No. patients	14	7
	Mean (SD)	10.2 (1.6)	8.5 (1.5)
	Median (range)	10.0 (7.5–13.0)	8.5 (7.0–11.1)
Foramen magnum area, mm^2^	No. patients	14	7
	Mean (SD)	119.4 (35.3)	97.2 (32.5)
	Median (range)	120.8 (64.4–163.8)	113.4 (49.6–129.9)
Myelon area, mm^2^	No. patients	14	7
	Mean (SD)	43.9 (19.4)	32.5 (12.0)
	Median (range)	38.4 (26.7–102.3)	31.4 (16.5–47.0)
Clivus length, mm	No. patients	16	7
	Mean (SD)	24.4 (3.2)	22.3 (2.2)
	Median (range)	24.3 (20.0–32.5)	22.0 (19.5–25.5)
Tentorium angle, degrees	No. patients	16	7
	Mean (SD)	47.7 (3.9)	46.9 (3.9)
	Median (range)	47.3 (38.0–56.5)	45.0 (43.5–54.0)
Occipital angle, degrees	No. patients	16	7
	Mean (SD)	78.7 (4.8)	76.4 (5.8)
	Median (range)	79.8 (70.5–85.0)	75.0 (70.0–85.0)
Posterior fossa volume, mm^3^	No. patients	10	3
	Mean (SD)	121,354.9 (33,762.1)	119,011.8 (34,362.1)
	Median (range)	127,663.5 (48,439.7–174,505.8)	106,081.3 (92,990.8–157,963.2)
Proportion of brainstem volume outside the posterior fossa, mm^3^	No. patients	10	3
	Mean (SD)	4227.0 (693.0)	4823.3 (1633.2)
	Median (range)	4448.4 (2778.6–5169.0)	4739.1 (3233.8–6497.0)
Cerebellum volume, mm^3^	No. patients	10	3
	Mean (SD)	95,285.9 (30,564.1)	93,927.2 (28,891.0)
	Median (range)	100,206.7 (29,592.2–141,281.0)	81,541.8 (73,293.8–126,946.0)
Supratentorial ventricular system volume, mm^3^	No. patients	14	7
	Mean (SD)	39,401.9 (24,497.0)	41,211.5 (15,093.3)
	Median (range)	33,539.9 (10,782.3–115,394.5)	37,651.3 (25,549.6–62,386.6)
Intracranial CSF system volume, mm^3^	No. patients	13	7
	Mean (SD)	229,855.0 (88,536.2)	201,651.5 (71,421.0)
	Median (range)	198,043.8 (118,999.2–376,599.0)	189,151.4 (124,900.2–337,691.4)
Fourth ventricle volume, mm^3^	No. patients	16	7
	Mean (SD)	831.5 (422.4)	651.5 (254.4)
	Median (range)	729.6 (338.3–1607.8)	635.7 (356.5–982.6)

## Data Availability

The data that support the findings of this study are available from the corresponding author, [KM], upon reasonable request.

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
