# Peer review of "Anatomical Characteristics of Cervicomedullary Compression on MRI Scans in Children with Achondroplasia"

_2313-433X, 2024, doi:10.3390/jimaging10110291_

Round 1
Reviewer 1 Report
Comments and Suggestions for Authors
Review of manuscript „Anatomical characteristics of cervicomedullary compression on MRI scans in children with achondroplasia“
General comments:
This manuscript presents findings from a retrospective study in children with achondroplasia, which is a non-lethal skeletal dysplasia. There exists a simple scoring system (Achondroplasia Foramen Magnum Score, AFMS) for categorizing the severity of the foramen magnum stenosis (FMS) based on magnetic resonance imaging (MRI ) scans. This score is useful for detecting spinal cord compression as early as possible to ensure timely intervention, e.g. by decompression surgery.
The aim of this study is to build upon the AFMS to identify further anatomical characteristics in children with achondroplasia. In total, 12 anatomical parameters were analyzed from 37 MRI scans of children with confirmed achondroplasia and compared with the parameters from 37 MRI scans of a reference group.
Of the 12 anatomical parameters, 10 showed significant differences between achondroplasia patients and reference group. These findings are of interest and deserve to be published. However, there are some issues that should be addressed in a revised manuscript. In particular, it does not become clear whether the additional parameters provide the basis for a more precise diagnosis when compared to the AFMS. In general, the discussion is relatively short.
In addition, the manuscript is intended for the section “Computer Vision and Pattern Recognition” in the special issue “Deep Learning in Computer Vision”. This does not seem to be appropriate, as mainly clinical findings and implications are discussed. No deep learning algorithms were developed or applied in this work.
Specific comments:
1. In this study, in total 12 anatomical parameters were assessed. It is not clear on what basis these specific parameters were selected. If they were selected based on findings from previous studies, the additional benefit from this study should be highlighted more precisely.
2. As stated in the discussion, the sample sizes of both achondroplasia and reference groups were small. Could you please discuss the potential and challenges for multicenter studies. For example, there exists studies with up to 49 patients. Pooling of several datasets could potentially useful.
3. Material and Methods: More information on the scan protocol (e.g. field strength, resolution, scanner settings such as echo time and repetition time) would be very helpful for reproducing the results.
4. Material and Methods: More information on the segmentation process would be helpful. It is not clear, if the segmenter software applied fully automated segmentation or required manual segmentation. Were the segmentations done by a single physician or several physicians?
5. In general, the reference list seems to be rather short.
6. Results / Figure 1d: the drawn tentorium angle is difficult to recognize.
Minor comments:
7. Results: “A total of 37 MRI images from 20 males …” may be rewritten as “A total of 37 MRI datasets from 20 males …” as there are more images per patient.
Author Response
Comments 1: General comments:
This manuscript presents findings from a retrospective study in children with achondroplasia, which is a non-lethal skeletal dysplasia. There exists a simple scoring system (Achondroplasia Foramen Magnum Score, AFMS) for categorizing the severity of the foramen magnum stenosis (FMS) based on magnetic resonance imaging (MRI ) scans. This score is useful for detecting spinal cord compression as early as possible to ensure timely intervention, e.g. by decompression surgery.
The aim of this study is to build upon the AFMS to identify further anatomical characteristics in children with achondroplasia. In total, 12 anatomical parameters were analyzed from 37 MRI scans of children with confirmed achondroplasia and compared with the parameters from 37 MRI scans of a reference group.
Of the 12 anatomical parameters, 10 showed significant differences between achondroplasia patients and reference group. These findings are of interest and deserve to be published. However, there are some issues that should be addressed in a revised manuscript. In particular, it does not become clear whether the additional parameters provide the basis for a more precise diagnosis when compared to the AFMS. In general, the discussion is relatively short.
Response 1: Thank you for your constructive feedback. In our revised manuscript, we have expanded the discussion to elaborate further upon the clinical relevance of our findings and their implications for future diagnosis and treatment of cervicomedullary compression in children with achondroplasia.
Comments 2: In addition, the manuscript is intended for the section “Computer Vision and Pattern Recognition” in the special issue “Deep Learning in Computer Vision”. This does not seem to be appropriate, as mainly clinical findings and implications are discussed. No deep learning algorithms were developed or applied in this work.
Response 2: In our revised manuscript we have highlighted in more detail that the anatomical landmarks identified provide an informative dataset to be used for the development of deep learning systems. We have also discussed the potential applications of such a system in (1) the indication of decompression surgery and (2) to measure the outcome following vosoritide or other precision therapies.
Comments 3: Specific comments
- In this study, in total 12 anatomical parameters were assessed. It is not clear on what basis these specific parameters were selected. If they were selected based on findings from previous studies, the additional benefit from this study should be highlighted more precisely.
Response 3: In our revised methods, we have elaborated on the reasons behind the selection of these 12 parameters. These parameters were selected from previous observational studies (Cohen et al. J Craniofac Genet Dev Biol Suppl. 1985;1:139-65; Calandrelli et al. Neuroradiology. 2017;59:1031−41) and based on personal experience of a board certified neurosurgeon. In our revised discussion, we also outline in greater detail the clinical relevance of our findings and their implications for future diagnosis and treatment of cervicomedullary compression in children with achondroplasia.
Comments 4: Specific comments
- As stated in the discussion, the sample sizes of both achondroplasia and reference groups were small. Could you please discuss the potential and challenges for multicenter studies. For example, there exists studies with up to 49 patients. Pooling of several datasets could potentially useful.
Response 4: Thank you. We have expanded the limitations section of our discussion to cover this point in greater detail.
Comments 5: Specific comments
- Material and Methods: More information on the scan protocol (e.g. field strength, resolution, scanner settings such as echo time and repetition time) would be very helpful for reproducing the results.
Response 5: The MRI images were collected from multiple institutions across Germany. Thus, there were likely subtle differences in the scan protocols employed between centers so we have not elaborated further upon this in our revised manuscript.
Comments 6: Specific comments
- Material and Methods: More information on the segmentation process would be helpful. It is not clear, if the segmenter software applied fully automated segmentation or required manual segmentation. Were the segmentations done by a single physician or several physicians?
Response 6: As requested, in our revised manuscript, we have expanded the methods section to include a more detailed description of the Segmenter software and its application in our study. Additionally, we have included images showing the segmentation process in our new Figure 2. The segmentations were completed by two physicians.
Comments 7: Specific comments
- In general, the reference list seems to be rather short.
Response 7: We have reviewed our introduction and methods, expanding both sections and citing a wider variety of previous literature.
Comments 8: Specific comments
- Results / Figure 1d: the drawn tentorium angle is difficult to recognize.
Response 8: In our revised manuscript we have included some revised images with clearer labeling of the tentorium angle.
Comments 9: Minor comments:
- Results: “A total of 37 MRI images from 20 males …” may be rewritten as “A total of 37 MRI datasets from 20 males …” as there are more images per patient.
Response 9: In our revised manuscript we have clarified that MRI images from 37 patients in total were analyzed. This included 20 males and 17 females.
Reviewer 2 Report
Comments and Suggestions for Authors
Strengths:
- Clinical Relevance:
The study tackles a significant issue in pediatric care, focusing on cervicomedullary compression in children with achondroplasia, a condition linked to serious complications like apnea and sudden infant death syndrome. This research holds potential for guiding clinical decisions related to early diagnosis and intervention. - Robust MRI Data:
The use of detailed MRI scans and measurement of 12 anatomical parameters provides valuable quantitative data for understanding anatomical differences. This can aid in early detection of foramen magnum stenosis (FMS) and inform decisions on decompression surgery. - Application of AFMS:
Retrospective application of the Achondroplasia Foramen Magnum Score (AFMS) enhances the clinical utility of the study by providing a standardized approach to determining the need for decompression surgery.
Major Concerns:
- Novelty and Contribution:
While the study is thorough, the novelty is limited as many of the anatomical differences, such as reduced foramen magnum size, are already well-established. The manuscript could benefit from clearly stating how it advances beyond previous literature, focusing on any novel anatomical findings or enhancements in the use of the AFMS. - Sample Size:
The sample size of 37 children with achondroplasia and 37 in the reference group is small, which may limit the generalizability of the findings. A power analysis is needed to ensure the sample size is adequate for detecting significant differences. Additionally, comparing children who underwent decompression surgery to those who did not may lack statistical power. - Mechanistic Insights:
The study provides anatomical data but lacks exploration of the mechanisms behind these abnormalities. More discussion on how specific findings, like a steeper tentorial angle or brainstem protrusion, contribute to clinical symptoms such as apnea would enhance the study's relevance to clinical practice. - Limitations and MRI Image Quality:
While limitations such as missing MRI sequences are mentioned, their impact on the findings is not thoroughly discussed. It would be helpful to explain how missing or incomplete data affected conclusions about anatomical structures and how these limitations were mitigated.
Minor Concerns:
- Methodology Clarification:
The rationale for selecting the 12 anatomical parameters should be explained more clearly, particularly in terms of their clinical significance. Additionally, elaboration on how measurement errors were controlled would improve methodological transparency. - Clinical Application Discussion:
The potential for MRI analysis to provide standardized guidelines for decompression surgery is mentioned but not fully explored. A more detailed discussion on how the findings could directly influence clinical thresholds for surgery would strengthen the study’s practical relevance. - Visual Aids:
The MRI images included in the study are useful but could be enhanced by labeling key structures and providing comparative images of children who did and did not undergo decompression surgery, highlighting the differences between these groups.
The quality of English is generally clear but could be improved in certain areas to enhance readability and clarity. Some sentences, particularly in the introduction and discussion sections, are complex and could benefit from simplification. Minor grammatical corrections and rephrasing would help make the manuscript more accessible to a broader audience. A final proofread focusing on sentence structure and flow would be beneficial.
Author Response
Comments 1: Strengths:
- Clinical Relevance:
The study tackles a significant issue in pediatric care, focusing on cervicomedullary compression in children with achondroplasia, a condition linked to serious complications like apnea and sudden infant death syndrome. This research holds potential for guiding clinical decisions related to early diagnosis and intervention. - Robust MRI Data:
The use of detailed MRI scans and measurement of 12 anatomical parameters provides valuable quantitative data for understanding anatomical differences. This can aid in early detection of foramen magnum stenosis (FMS) and inform decisions on decompression surgery. - Application of AFMS:
Retrospective application of the Achondroplasia Foramen Magnum Score (AFMS) enhances the clinical utility of the study by providing a standardized approach to determining the need for decompression surgery.
Major Concerns:
- Novelty and Contribution:
While the study is thorough, the novelty is limited as many of the anatomical differences, such as reduced foramen magnum size, are already well-established. The manuscript could benefit from clearly stating how it advances beyond previous literature, focusing on any novel anatomical findings or enhancements in the use of the AFMS.
Response 1: In our revised manuscript, we have expanded the discussion to elaborate further upon the clinical relevance of our findings and their implications for future diagnosis and treatment of cervicomedullary compression in children with achondroplasia.
Comments 2:
- Sample Size:
The sample size of 37 children with achondroplasia and 37 in the reference group is small, which may limit the generalizability of the findings. A power analysis is needed to ensure the sample size is adequate for detecting significant differences. Additionally, comparing children who underwent decompression surgery to those who did not may lack statistical power.
Response 2: Thank you for this comment. This was an exploratory study and as such, a power analysis was not conducted. In our revised manuscript we have included an expanded discussion of the limitations of our analysis, which we hope is acceptable.
Comments 3:
- Mechanistic Insights:
The study provides anatomical data but lacks exploration of the mechanisms behind these abnormalities. More discussion on how specific findings, like a steeper tentorial angle or brainstem protrusion, contribute to clinical symptoms such as apnea would enhance the study's relevance to clinical practice.
Response 3: In our revised manuscript, we have expanded the discussion to elaborate further upon the clinical relevance of our findings and their implications for future diagnosis and treatment of cervicomedullary compression in children with achondroplasia.
Comments 4:
- Limitations and MRI Image Quality:
While limitations such as missing MRI sequences are mentioned, their impact on the findings is not thoroughly discussed. It would be helpful to explain how missing or incomplete data affected conclusions about anatomical structures and how these limitations were mitigated.
Response 4: Thank you. We have expanded the limitations section of our discussion to cover this topic in greater detail.
Comments 5: Minor Concerns:
- Methodology Clarification:
The rationale for selecting the 12 anatomical parameters should be explained more clearly, particularly in terms of their clinical significance. Additionally, elaboration on how measurement errors were controlled would improve methodological transparency.
Response 5: In our revised methods, we have elaborated on the reasons behind the selection of these 12 parameters. These parameters were selected from previous observational studies (Cohen et al. J Craniofac Genet Dev Biol Suppl. 1985;1:139-65; Calandrelli et al. Neuroradiology. 2017;59:1031−41) and based on personal experience of a board certified neurosurgeon. In our revised discussion we highlight that each measurement was conducted in duplicate to compensate for any measurement errors.
Comments 6:
- Clinical Application Discussion:
The potential for MRI analysis to provide standardized guidelines for decompression surgery is mentioned but not fully explored. A more detailed discussion on how the findings could directly influence clinical thresholds for surgery would strengthen the study’s practical relevance.
Response 6: In our revised manuscript, we have expanded the discussion to elaborate further upon the clinical relevance of our findings and their implications for future diagnosis and treatment of cervicomedullary compression in children with achondroplasia.
Comments 7:
- Visual Aids:
The MRI images included in the study are useful but could be enhanced by labeling key structures and providing comparative images of children who did and did not undergo decompression surgery, highlighting the differences between these groups.
Response 7: In our revised manuscript, we have included new Figures 1 and 2, which now show each of the 12 parameters measured. The figures show MRI scans from a patient who did not undergo decompression surgery. We have refrained from including representative images from a patient who did undergo decompression surgery and have instead included images illustrating the segmentation process in response to comments from Reviewer 1, which we hope is acceptable.
Comments 8: Comments on the Quality of English Language
The quality of English is generally clear but could be improved in certain areas to enhance readability and clarity. Some sentences, particularly in the introduction and discussion sections, are complex and could benefit from simplification. Minor grammatical corrections and rephrasing would help make the manuscript more accessible to a broader audience. A final proofread focusing on sentence structure and flow would be beneficial.
Response 8: We have reviewed our revised manuscript and simplified the wording where possible to ensure it is clear and understandable for all audiences.
Round 2
Reviewer 2 Report
Comments and Suggestions for Authors
Thank you for your detailed response and the effort you have put into revising the manuscript. I appreciate the comprehensive adjustments made, which have contributed to a clearer presentation of the findings.